# Applications of Antimicrobial Photodynamic Therapy against Bacterial Biofilms

**DOI:** 10.3390/ijms23063209

**Published:** 2022-03-16

**Authors:** Sandile Phinda Songca, Yaw Adjei

**Affiliations:** 1School of Chemistry and Physics, College of Agriculture Engineering and Science, University of KwaZulu-Natal, Durban 4041, South Africa; 2Water Research Institute, Council for Scientific and Industrial Research, Accra 8164, Ghana; anadjei@yahoo.com

**Keywords:** biofilm, planktonic bacteria, extracellular polymeric substance, antimicrobial photodynamic therapy, antibiotic chemotherapy, photothermal hyperthermia therapy, magnetic hyperthermia therapy, cold atmospheric pressure plasma, sonodynamic therapy, nanozyme enhanced photodynamic therapy

## Abstract

Antimicrobial photodynamic therapy and allied photodynamic antimicrobial chemotherapy have shown remarkable activity against bacterial pathogens in both planktonic and biofilm forms. There has been little or no resistance development against antimicrobial photodynamic therapy. Furthermore, recent developments in therapies that involve antimicrobial photodynamic therapy in combination with photothermal hyperthermia therapy, magnetic hyperthermia therapy, antibiotic chemotherapy and cold atmospheric pressure plasma therapy have shown additive and synergistic enhancement of its efficacy. This paper reviews applications of antimicrobial photodynamic therapy and non-invasive combination therapies often used with it, including sonodynamic therapy and nanozyme enhanced photodynamic therapy. The antimicrobial and antibiofilm mechanisms are discussed. This review proposes that these technologies have a great potential to overcome the bacterial resistance associated with bacterial biofilm formation.

## 1. Introduction

Originally discovered as a relatively new anticancer therapeutic technology [1], photodynamic therapy (PDT) has since evolved and is now used in many therapeutic technologies. For example, it is used against viruses, such as the recently reported antiviral activity against COVID-19 [2], bacteria [3], fungi [4] and parasites [5], in treating neovascular disease [6], in environmental sanitation [7] and pest control [8], and in many other applications. Besides anticancer applications, photodynamic therapeutic applications against bacteria have defined the knowledge field of antimicrobial photodynamic therapy (aPDT) [9] and the allied healthcare variant known as photodynamic antimicrobial chemotherapy [10]. Common reference to both photodynamic antimicrobial chemotherapy and photoactivated chemotherapy (PACT) as antimicrobial chemotherapy is unfortunate because oxygen in the latter is not required [11]. PDT has been shown to kill many bacterial species in planktonic and biofilm formations [12,13]. Although aPDT photosensitizers have shown very little or no tendency to induce it [14,15,16], some mechanisms of bacterial resistance have been reported, including ABCG2 mediated efflux, DNA damage repair, procaspase damage-induced inhibition of apoptosis, heat shock protein upregulation, hypoxia, and antioxidant defense mechanisms [17]. Given that biofilm formation is one of the most important mechanisms for the development of bacterial resistance, there has been increasing focus on the development of new methods for the treatment of bacterial biofilms [18], among which PDT has been highly prized with an increasing number of research reports showing promising results in vitro and in vivo [19,20]. 

Furthermore, aPDT has been used in combination with several other therapeutic agents, with additive and synergistic efficacy enhancement [21,22]. aPDT combinations with chemotherapy have been studied for the potential treatment of bacterial infections [23]. Other combination therapy studies include PDT with anticancer chemotherapy [24], photothermal hyperthermia therapy (PTT) for anticancer applications [25], antibacterial applications [26], and antifungal applications [27]. PDT has also been studied in combination with magnetic hyperthermia therapy (MHT) for various applications including hard-to-reach cancers such as brain cancer [28] and bone tissue cancers [29]. As a therapeutic technology, MHT involves the temperature elevation of tissues in which embedded magnetic nanoparticles are energized using a high frequency alternating magnetic field generated by an appropriate MHT applicator [30].

When combined with cold atmospheric pressure plasma therapy (CAP), PDT has the potential to overcome hypoxia, which limits the concentration of reactive oxygen species that can be generated by PDT [31,32]. The reason for this is that CAP essentially introduces exogenous reactive gas species from a plasma jet (in the case of the direct CAP) or from a plasma activated fluid (in the case of the indirect CAP). When a gas such as helium or oxygen is passed over a high voltage electrode, reactive gas species are generated at room temperature in the gas phase, including high energy electrons and gas phase ionic species, creating a cold atmospheric pressure plasma gas as a controllable source of reactive species [33]. These reactive species are widely used to destroy undesirable cells directly or indirectly. Devices used in CAP include the dielectric barrier discharge device [34] PlasmaDerm^®^ VU-2010 [35], the atmospheric pressure plasma jet kINPen^®^ MED [36], and the SteriPlas [37], which are CE-certified as medical products to treat chronic wounds in humans with efficacy and a good tolerability. 

Yan et al. (2017) further distinguishes between the two approaches used to generate cold atmospheric pressure plasma, namely the direct and indirect discharges, upon which the plasma jet and the dielectric barrier discharge devices are based [38]. Other anticancer combinations include PDT with radiotherapy [39] and immunotherapy [40,41]. This review, however, focuses on the applications of PDT and several combinations thereof against bacterial and fungal biofilms. As a background, this review explores how biofilms are formed and the various strategies that are used against biofilms before delving deeper into the applications of PDT and combination therapies thereof. Nanotechnology is a constant theme in the discussions of the applications of the various combinations of PDT.

## 2. The Microbial Biofilm Structural Challenge

Deeper understanding of the biofilm structure and how it contributes to antibiotic resistance has improved since the late 1970s [42], when they were first recognized in clinical samples. Biofilms are constituted from microbial communities that intimately associate with surfaces to sustain viability and improve their resistance up to a thousand times more than their planktonic forms; the surface contact and association is a key requirement for the biofilm formation mechanism [43]. Biofilms are characterized by an extensive multi-channel-permeated extracellular 3D matrix among the cells, consisting of polymeric materials, including polysaccharides and peptides, nucleic acids and lipids [44]. The biomolecule constitution of the biofilm matrix and extracellular polymeric substances (EPS) has been described in more detail [45]. It plays a vital role as a living environment for the bacterial cells in the biofilm [46] and as an infective mechanism through detachment and re-attachment [47]. Researchers have recognized several stages in biofilm formation, including contact and adhesion, formation of the colony, biofilm architecture maturation, final detachment and distal infection. 

Therefore, bacterial biofilms enhance antibiotic resistance and further infection [48]. Furthermore, the formation of biofilm is a collective behavior that is triggered by cell-to-cell proximity and facilitated by a bacterial quorum-sensing mechanism [49]. Due to their importance for therapeutic strategy formulation, several studies on the physical, chemical and mechanical properties of biofilms such as substrate adhesion, adsorption of chemical substances, and admission of other cells, have been conducted [50,51,52,53]. These studies have revealed that antibiotics acting on their own do not alter the biofilm structure to gain sufficient access to the constituent bacterial community. Many of these properties can be exploited to overcome the formidable bacterial biofilm formation defense mechanism. 

For example, their sessile nature may permit therapeutic targeting and imaging for precision drug delivery. The bacterial biofilm is a hydrophilic and wettable environment, amenable to aqueous soluble drugs and delivery strategies. Variable oxygenation levels have been reported, and the multi-channel 3D structure and reported hypoxia suggest limited susceptibility to the type II mechanism of PDT [54]. The chemical composition of the EPS of the biofilm may be among the key considerations for drug design and targeting strategies, because drugs and photosensitizers that bind to any of the known constituents of the EPS biofilm matrix could contribute toward biofilm structure disruption [55]. For these reasons, the affinity for biofilm penetration may be a key factor in therapeutic strategies.

## 3. Antimicrobial Photodynamic Therapy

In the aPDT approach, absorbed light energy is always used for bactericidal or bacteriostatic impact through two key molecular photosensitizer-mediated mechanisms. While the type I mechanism is based on radical-forming hydrogen transfer from a light energized photosensitizer directly to biomolecules, the type II mechanism involves an initial photosensitization of oxygen to produce reactive oxygen species, which in turn attack biomolecules [56]. The mechanistic basics of type I and type II may be illustrated using a Jablonski diagram as shown in Figure 1, showing the direct and the reactive oxygen species mediated pathways. Studies have shown that both mechanisms cause irreversible chemical reactions that alter the functionality of biomolecules and disrupt their environment [57], regardless of whether these biomolecules are cellular, EPS matrix components or other functional constituents of the biofilm [58]. These studies showed that aPDT is not prevented by the biofilm from achieving an increase in intracellular reactive oxygen species. At the same time, several studies have focused on the impact of aPDT on the biofilm matrix strength and constituent pathogen metabolic activity [59,60]. For example, bacterial pathogen reduction and eradication was found to be accompanied by reactive oxygen- induced oxidative stress, biofilm matrix weakening, loss of adhesion and component changes [61]. 

Several aPDT studies have been conducted on the planktonic and biofilm forms of bacterial and fungal pathogens. Using aPDT for example, the planktonic and polymicrobial biofilms of methicillin-resistant *Staphylococcus aureus*, *Pseudomonas aeruginosa* and associated fungi were reduced to less than 99.99% using methylene blue as the photosensitizer and 670 nm laser light for excitation of the photosensitizer [62]. Additionally, aPDT completely eradicated Gram-negative *Moraxella catarrhalis* and Gram-positive *Streptococcus pneumoniae* in both their planktonic and biofilm forms using the photosensitizer Chlorin-e6 and 670 nm light for excitation of the photosensitizer [63]. Biofilm-forming *Acinetobacter baumannii* planktonic cells and biofilms were eradicated using methylene blue and protoporphyrin IX photosensitizers, illuminated at 652 nm for excitation of the photosensitizer [64]. These studies show that aPDT is effective in eradicating biofilm-forming bacterial strains in both their planktonic and biofilm forms. Table 1 summarizes the methodologies, pathogens and impact of the six key studies discussed.

## 4. Problem Statement

The formation of biofilms is one of the mechanisms for the development of bacterial resistance among most bacteria [65]. It protects bacteria from host immune defenses and antibiotics [66]. Most aPDT study methodologies involve preincubation of bacteria with the photosensitizer to effect retention by the biofilm matrix, or the bacterial cell walls, before irradiation. Such retention may be due to bacterial binding to the biofilm EPS matrix, cell wall or bacterial intracellular uptake and retention. Studies have shown that these are essential for the method, and without preincubation of the bacteria with the photosensitizer, aPDT is not effective [67]. Some of the commonly available photosensitizers have been reported to be ineffective as aPDT agents against especially Gram-negative bacteria. For example, toluidine blue was not effective against *Staphylococcus aureus* [68]. 

Antibiotics have been the main weapon in the fight against bacterial infections for close to 90 years. In recent times, however, the effectiveness of antibiotics has been severely compromised by the rising incidence of antibiotic resistance, which has rendered many antibiotics ineffective against bacterial infections, notably those forming bacterial biofilms. The World Health Organization prediction that ~10 million people could die every year around the world by 2050 if the current trend in antibiotic resistance remains unchanged suggests that antibiotic resistance is a crisis the world is facing today [69]. Therefore, the observed increase in aPDT research, especially aimed at biofilms, could be among the timely interventions to avert this crisis [70]. 

## 5. Combination Therapies with Antimicrobial Photodynamic Therapy

PDT has been used in combination with several non-invasive therapeutic approaches with additive and synergistic efficacy and enhancement of outcomes in most studies conducted in vitro, as well as in preclinical and clinical applications [71]. In this regard, studies and applications have been reported for aPDT in combination with antibiotic chemotherapy [72] PTT [73], MHT [74], CAP [75,76], and endodontic debridement [77]. Studies and applications have also been reported in combinations using multiple PDT photosensitizers, in what could be termed multiple photosensitizer combination aPDT. In many multiple photosensitizer combination studies, inorganic-organic [78] and organic-inorganic [79,80] hybrid photosensitizers are recognized. 

Some combination therapy studies that include aPDT against antibiotic resistant biofilm-forming bacteria have been conducted directly without the use of nanomaterials [81]. However, a constant theme found in most of the strategies used for combination therapies involving aPDT is the application of nanomaterials [82]. In these strategies, the nanomaterials are used for various purposes: as agents for transport and delivery, disease site targeting, microbial cell specificity, and the release of combination therapy agents in response to external stimuli, or to both the external and internal microbial cell microenvironment. In a tangential combination, Hamblin and Abrahamse reported a remarkable enhancement of the antimicrobial photodynamic effect by the addition of aqueous solutions of inorganic salts against planktonic bacteria [83] and potentially against biofilms [84]. The foregoing includes some of the considerations that inspired the current reflections on the applications of combinations of aPDT against biofilms presented in this review. 

While many of the combination therapies are still in experimental studies in vitro and preclinical studies in vivo, quite a few have progressed to clinical trials with a significant number of reports and clinical case study communications. Therefore, there is a translation pipeline from basic studies to clinical applications in many of the studied applications of combination therapies. For example, several clinical trials of the combination of antibacterial chemotherapy in combination with PDT have been reported for the treatment of periodontal disease [85,86]. Evidence that aPDT in combination with antimicrobial chemotherapy is a successful therapy in clinical practice and that its usefulness as a clinical treatment for bacterial infections is being recognized by clinicians to have very good clinical prospects is provided by a clinical case study of atypical mycobacterial skin infections, a rare type of refractory infection [87]. The aim of the study was to evaluate the efficacy and safety of 5-aminolevulinic acid-mediated aPDT combined with several antibiotics (moxifloxacin, clarithromycin, amikacin, imipenem mixed with cilastatin, rifampicin, ethambutol, and levofloxacin) in the combination treatment of *Mycobacterium abscessus*, *gordonae*, *gilvum*, and *fortuitum* skin infections. In agreement with a separate case study of the treatment of *Mycobacterium fortuitum* skin abscesses, all enrolled patients were cured with 100% efficiency [88]. 

## 6. Nanoparticle-Photosensitizer Conjugate

By far the most widely used strategies for studying the effects of aPDT against biofilms use nanoparticle-photosensitizer conjugates that are typically engineered and fabricated to incorporate the photosensitizer into the conjugate in such a way that it retains its activity in the photodynamic reaction. There are several nanoparticulate materials that act as PDT photosensitizers on their own. These include nanoparticles of copper sulphide [89], zinc oxide [90], iron oxide [91], silver [92], gold [93], nano graphene oxide [94], porphyrins [95] and phthalocyanines [96]. All of the foregoing nanoparticles have been used as photosensitizers in many studies of the application of aPDT against biofilms. For example, nanoparticles of zinc oxide loaded onto zeolite framework showed high singlet oxygen quantum yields, biofilm matrix compromise, bactericidal effects, and excellent remineralization following extensive microbial demineralization [90]. Several studies have shown that nanoparticles of silver possess sufficient antibiofilm and antibacterial properties to inhibit the formation of biofilms and eradicate both biofilms and embedded microbiota such as *Pseudomonas aeruginosa*, *Escherichia coli*, and *Staphylococcus aureus* [97,98]. Due to the photothermal and magnetothermal conversion capability of some of these nanomaterials, they are also used as the nanomaterial agents for combination therapies involving photothermal, magnetothermal, and PDT. For example, nanographene oxide and copper sulfide nanoparticles are used as photothermal and PDT agents [89,94], while iron oxide nanoparticles [91] are of use in magnetothermal, photothermal and PDT combinations. 

In a study of self-assembled photosensitizers, a nanoemulsion encapsulated cationic chloro-aluminum phthalocyanine reduced the biofilm metabolic activity by 80% and 73% for the methicillin-susceptible and methicillin-resistant *Staphylococcus aureus* suspensions and biofilms, respectively, eradicating both bacterial strains, whereas the anionic counterparts were not as effective [99]. Like methylene blue capped silver nanoparticles [100], methylene blue capped gold nanoparticles were effective in eradicating *Candida albicans* planktonic cells and biofilm populations [101]. Mesoporous silica nanoparticles loaded with malachite green also eradicated *Staphylococcus aureus* and *Escherichia coli* planktonic cells and biofilms [102], reducing the metabolic activity by 69%. Chlorin-e6 conjugated manganese oxide nanosheets assembled to form a pH responsive nanoconjugate by means of bovine serum albumin and polyethylene glycol were reported to significantly reduce biofilm formation by aPDT and eradicate the bacterial population [103]. Responsiveness to biofilm microenvironmental characteristics such as acidity, hypoxia, enzyme and hydrogen peroxide concentration has been exploited to trigger nanomaterial-based photosensitizer and chemotherapy drug delivery and release, and to enhance targeting of disease sites and cells [104,105].

A nanoconjugate photosensitizer formed by conjugation of indocyanine green with graphene oxide nanodots showed a remarkable reduction of biofilm forming by *Enterococcus faecalis*, along with a reduction in the viability and integrity of the biofilms, following aPDT using 200 micrograms of the nanoconjugate per milliliter [106]. The foregoing sample of research studies suggests that nanoparticle-mediated aPDT represents a viable alternative for the eradication of bacterial biofilms and therefore a major strategy to combat biofilm based antibiotic resistance. 

## 7. Combination with Antibiotic Chemotherapy

It has been noted that some combination therapy studies of aPDT with the use of antibiotic chemotherapy agents against antibiotic resistant biofilm-forming bacteria have been conducted directly without the use of nanomaterials. For example, aPDT using indocyanine green and ethylenediaminetetraacetic acid in combination with the antibiotic chemotherapy agents vancomycin, minocycline, and cefepime showed significant synergistically enhanced efficacy and disruption of the biofilm structure of methicillin resistant *Staphylococcus aureus* and *Pseudomonas aeruginosa* [107]. In the study, susceptibility measurements were conducted using the disc diffusion method, and the viability of the bacteria was evaluated using the minimum bacterial concentration. Bacterial metabolic activity reduction evaluated by the resazurin assay aligned well with the extent of disruption of the biofilm, which was clearly shown using confocal laser scanning microscopy. Five key combinations of aPDT with antibiotic chemotherapy are summarized in Table 2. 

On the other hand, most studies have shown that nanoconjugate-mediated aPDT in combination with antibiotic drug chemotherapy is quite effective in bacterial biofilm disruption. For example, photodynamic treatment with amoxicillin-coated gold nanoparticles, in which the nanogold acted as the photosensitizer and the amoxicillin was the antibiotic agent, penetrated the biofilms, eradicating the embedded *Pseudomonas aeruginosa* and *Staphylococcus aureus* [108]. Upon treatment of the biofilms of *Escherichia coli*, *Staphylococcus aureus*, and methicillin-resistant *Staphylococcus aureus* with a zeolitic imidazolate framework-8-polyacrylic acid loaded with methylbenzene blue as the photosensitizer and vancomycin as the antibiotic drug, the biofilm matrix structure was compromised, allowing sufficient penetration by the nanoconjugate and eradication of the bacteria [109]. The nanoconjugate confers responsive drug release, triggered by the pH of the external environment of the biofilm and the internal environment of the bacteria, while allowing for loading of large quantities of the methylene blue photosensitizer. It is also coated with amino-functionalized polyethylene glycol for loading of large quantities of the vancomycin antibiotic chemotherapy agent. The significance of nanoparticle-mediated combination therapy studies involving antibiotic chemotherapy drugs with aPDT agents against biofilm-forming pathogens is the potential to enhance the therapeutic effects of antibiotic chemotherapy drugs by reducing their potential to induce the development of antibiotic resistance. This could be used to restore many antibiotic chemotherapy drugs rendered ineffective by bacterial resistance and repurpose them to useful applications [112].

In addition to the many case studies reported on the application of this combination therapeutic technology in periodontal therapy [87,88], enhancement of healing effects has also been demonstrated with the healing of third degree burn wounds infected with methicillin-resistant *Staphylococcus aureus*, *Escherichia coli*, and *Pseudomonas aeruginosa* in mice, using protoporphyrin IX as the aPDT photosensitizer and ceftriaxone as the antibiotic drug [110]. Furthermore, a clinical comparative study demonstrated improvement in clinical and histological outcomes of aPDT using indocyanine green as the photosensitizer, combined with antibiotic therapy using amoxicillin as the antibiotic drug, for pericoronitis treatment when compared with antibiotic therapy alone [111]. 

## 8. Combination with Photothermal Hyperthermia Therapy

PTT is a therapeutic technology in which plasmonic nanomaterials are used as photothermal conversion agents to elevate the temperature of disease tissues or cells in which they are embedded [113]. Several experimental studies have reported synergistic enhancement of antibiofilm and bactericidal effects resulting from bringing together aPDT and PTT directly to eradicate planktonic and biofilm formations of virulent bacterial pathogens. To illustrate this approach with an example, an in vitro study of toluidine blue mediated aPDT in combination with the indocyanine green mediated PTT revealed that the combination significantly reduced *Streptococcus mutans* colony forming units with more pronounced inhibition of biofilm formation compared to the control upon irradiation with a diode array laser at 635 nm [114]. The combination therapy was achieved by directly combining the PTT and the aPDT agents without incorporating them in a nanoconjugate system. Studies, however, are increasingly based on nanomaterials as the carriers of the photosensitizer and the photothermal conversion agents. To illustrate the nanomaterial-mediated strategy with an example, a photothermal-aPDT combination based on indocyanine green loaded aminopropyl silane capped superparamagnetic iron oxide nanoparticles achieved several log reductions of Gram-negative *Escherichia coli*, *Klebsiella pneumoniae*, *Pseudomonas aeruginosa*, and Gram-positive *Staphylococcus epidermis* planktonic cells in vitro, with complete biofilm eradication [26]. Similarly, a combination of the photothermal-antimicrobial photodynamic approach based on indocyanine green loaded mesoporous polydopamine nanoparticles functionalized with arginylglycylaspartic acid synergistically eradicated the *Staphylococcus aureus* biofilm and all the embedded bacterial cells found in titanium surgical bone-implants in vivo [114]. The nanomaterial-mediated photothermal-aPDT methodology has been modified by numerous researchers to overcome many of its limitations. For example, an indocyanine green and manganese pentacarbonyl bromide-doped dendrimer-based nanogel generated sufficient quantities of carbon monoxide to overcome collateral tissue damage and inflammation in the photothermal-photodynamic eradication of *Escherichia coli* and methicillin-resistant *Staphylococcus aureus* [115]. Environmentally responsive releases of chemotherapy drugs and photosensitizers endow these studies with high specificity. For example, a polymer nanoconjugate platform with a diketopyrrolopyrrole-based photothermal agent exhibited lipase-triggered release of incorporated triconazole and fluconazole, eradicating *Candida albicans* biofilms and planktonic cells with a high degree of specificity [116]. Five studies that illustrate the combination of aDPT and PTT are listed in Table 3. 

The proliferation of nanomaterial-mediated combinations of aPDT and PTT may be attributed to the outcomes of these investigations, in which such synergistic enhancement is more pronounced. As a result, several research groups are exploring how nanomaterial-based combination therapies in general can be taken further than the ubiquitously reported dual combinations, to triple combinations and possibly beyond. To illustrate this with an example, a triple therapy combination of aPDT with PTT and nanozyme reactive oxygen species generation was reported recently to achieve temperature elevation-modulated and reactive oxygen species-mediated broad-spectrum sterilization of multi-pathogenic biofilms in an environment that closely resembles those found in burn wounds [119]. The enhanced production of reactive oxygen species derives from the combined photodynamic production of singlet oxygen and the nanozyme generation of hydroxyl radicals, both of which are due to oxygen vacancies on the surface of the molybdenum trioxide nanozyme [120] as shown in Figure 2. 

The method produced complete closure of 1 cm diameter wounds in six days compared to the controls, in which the wounds were still more than 60% open during this period. A nanoplatform for another triple combination therapy involving silver nanoparticle-based chemotherapy, indocyanine green-based aPDT and molybdenum disulphide-based PTT was fabricated by decorating the molybdenum disulphide nano-sheets with indocyanine green and silver nanoparticles. This therapy also showed broad-spectrum sterilization and biofilm structural destruction reaching deep into the biofilm [121], closing 1 cm wounds in 7 days, compared to the controls, in which the wounds did not reach 50% closure during this period. These molybdenum disulphide-based, and other triple therapy proofs-of-concept have been on the rise over the past few years [122,123]. Three typifying tritherapy combinations involving aPDT, PTT, and nanozyme activity are listed in Table 4. 

High aspect ratio gold nanorods are known for their high photothermal conversion and ablation of cancer cells [124], bacteria and biofilms in vitro [125]. Basic research studies have illustrated the conceptual simplicity of putting together a photothermal-photodynamic combination therapy strategy by conjugating gold nanorods with a high singlet oxygen quantum yield photosensitizer such as toluidine blue [117]. Therefore, it must be questioned why the photothermal-photodynamic combination therapy has yet to reach clinical applications. Wei et al. (2020) identify poor light penetration of tissue even in the therapeutic near infrared window as one of the reasons for the paucity of clinical trials and clinical case reports, which suggests that the combination of photothermal and PDT will be limited to low depth skin and wound infections [126], even though pre-clinical studies are increasingly demonstrating enhanced wound healing effects [118,122,123]. It is also possible that serial or simultaneous irradiation of the photosensitizer and the photothermal conversion agent raises the cost of the technology by introducing the need for two irradiation wavelength light sources [127].

## 9. Combination with Magnetic Hyperthermia Therapy

The use of MHT has been reported to reduce viability of biofilm-forming bacteria, compromise the EPS matrix of their biofilms [128,129,130,131], and induce the innate immune response [132]. For these reasons, it has been used in combination with antibiotic chemotherapy, in which disruption of the biofilm enables the antibiotic to access the pathogen cells [133], and has also been used for the treatment of biofilms growing on the surface of surgical implants and prosthetics [134]. However, to the extent of our literature search, no studies of MHT in combination with PDT have been reported. When an external magnetic field was used for targeting of chlorin e6-laden and mesoporous silica-capped iron oxide magnetic nanoparticles during aPDT [135], it caused the magnetic nanoparticles to move deep into the biofilm, without an alternating magnetic field as in MHT [136]. A nanoconjugate consisting of superparamagnetic iron oxide nanoparticles capped with the photosensitizer curcumin showed magneto-thermal conversion upon application of an alternating magnetic field and excellent PDT effects upon irradiation with blue light, eradicating planktonic *Staphylococcus aureus*. Surprisingly however, no experiments were conducted on the combination of MHT with PDT using this nanoconjugate in this study [74]. These comparative studies of aPDT and MHT are listed in Table 5. 

The apparent absence of combination studies using MHT and PDT for the eradication of biofilms and biofilm-forming microorganisms is puzzling, given that applications of the innovative combination against cancer have been widely reported [137]. However, studies have shown that while both direct and high frequency alternating external magnetic field achieve disruption of the biofilm matrix, more extensive biofilm matrix damage is obtained with the direct magnetic field compared with the high frequency alternating magnetic field [118]. This low appetite for the combination of aPDT and MHT may be attributed to the potentially high capital costs of investment for low therapeutic returns, given that clinical MHT applicators such as the patented MFH^®^300F high frequency alternating magnetic field applicator for humans are relatively new, few and therefore very expensive [138]. 

The mechanism for the combination of MHT with PDT envisages the same design and application of a suitable photosensitizer-loaded nanoconjugate used for anticancer applications [106]. Application of the nanoconjugate to the biofilm followed by temperature elevation caused by the MHT would lead to the release of the photosensitizer from the nanoconjugate and weakening of the EPS matrix. Simultaneous or serial irradiation would lead to unleashing of reactive oxygen species and radicals through type I and II photosensitizer reactions on the biofilm and embedded pathogens. The combined effect of the temperature elevation, weakening of the EPS matrix, and proliferation of radical and reactive oxygen species would be to eradicate both the biofilm and biofilm forming pathogens embedded in the biofilm. To the extent of our search of the published literature, this experiment has not been conducted. 

## 10. Combination with Cold Atmospheric Pressure Plasma Therapy

Several research studies on the applications of CAP against bacterial infection have shown good results over the past two decades [139,140]. The comparison of the effects of aPDT with those of CAP on biofilm-forming bacteria and the biofilms they form has been widely studied, in most cases showing very good biofilm and bacterial eradication, comparable to the results obtained from aPDT studies. For example, several research groups compared the relative performance of CAP in the eradication of bacterial biofilms and embedded bacteria with aPDT [76,141,142,143,144]. The results of these comparative studies showed that although CAP significantly reduced bacterial biofilms and inhibited their formation, the overall relative performance was in par or less than that of aPDT, in line with conclusions made by an earlier comparative review [145]. Therefore, the researchers recommended further improvement of the CAP combination with aPDT. It is therefore conceivable that a combination of CAP with aPDT emerges as a major contribution to antimicrobial warfare. The combination of CAP with aPDT in clinical treatment of septic wounds, for example, would typically involve the application of one of the cold atmospheric pressure plasma devices such as the dielectric barrier discharge to generate the plasma jet [34,35,36,37]. The plasma jet produces sufficient light for the aPDT production of reactive oxygen species, including singlet oxygen and hydroxyl radicals. Table 6 summarizes five key studies of CAP combined with aPDT. 

Indeed, recent reports of the combination of CAP with aPDT confirming the above expectations can be supported by a recently published doctoral degree study, still appearing as the original thesis in its institutional library. The doctoral candidate studied the synergistic effects of CAP in combination with PDT in which Rose Bengal was used as the photosensitizer [75]. The research presents much evidence to support the view that the combination of CAP with aPDT holds great potential as a successful new approach for healing wounds. However, given that a report emanating from this work has not yet been published and therefore accordingly peer-reviewed, it may be early to comment on the evidence, even though the thesis was duly examined by two referees. To the extent of the search of the published literature conducted by us and by the doctoral candidate, this is the first study conducted to evaluate the effects of aPDT in combination with CAP. The study approach was to expose the wounds to aPDT, followed by the application of the cold atmospheric pressure plasma from a dielectric barrier discharge device. The overall conclusion of the doctoral degree study is that the combination of cold atmospheric pressure plasma and aPDT synergistically enhanced the bactericidal and biofilm eradication effects of either techniques acting alone, and therefore presents a promising method.

## 11. Combination with Sonodynamic Therapy

Sonodynamic therapy (SDT) is an innovative combination of PDT and ultrasound that involves exposing diseased tissues to chemical compounds that produce reactive oxygen species upon sensitization by means of low-intensity ultrasound. In addition to the reported anticancer research applications [146], it has been applied to antimicrobial research studies with promising results in vitro [147,148,149]. The evaluation of SDT may be illustrated with the comparison between PDT and SDT for *Staphylococcus aureus* biofilm samples [150]. The research compared ultrasound treatment, photodynamic treatment and the combined ultrasound and photodynamic treatment, and found that the combined treatment reduced the planktonic and biofilm of *Staphylococcus aureus* more than ultrasound treatment alone, and more than photodynamic treatment alone. As correctly pointed out by Fan et al. (2021), after many mechanistic and efficacy studies in vivo, clinical SDT is imminent [150].

Another example involved the design and fabrication of a nanoconjugate of titanium dioxide loaded with sinoporphyrin sodium (Figure 3a), as the photosensitizer. Eradication of planktonic *Staphylococcus aureus* and biofilm upon exposure to low-intensity ultrasound treatment using the nanoconjugate exceeded that obtained from the same treatment using titanium oxide and that obtained using sinoporphyrin sodium [151]. A variant of this nanoconjugate consisting of protoporphyrin IX (Figure 3b)-laden mesoporous nanosilica, in which the nanoconjugate surface-anchored ferrous ions were used for the Fenton reaction production of hydroxyl radicals and perhydroxyl radicals (Figure 4 and Equation (3)) to enhance the reactive oxygen species production, reduced planktonic *Enterococcus faecalis* and biofilms with high efficiency [152].
Fe^2+^ + H_2_O_2_ → Fe^3+^ + •OH + OH^−^(1)
Fe^3+^ + H_2_O_2_ → Fe^2+^ + •OOH + H^+^(2)
2H_2_O_2_ → •OOH + •OH + H_2_O(3)

One advantage of antimicrobial SDT over aPDT is the deeper tissue penetration of ultrasound waves, which enables effective reach into deeper lying disease through poorly light-penetrable tissues such as bones and teeth [152]. The efficacy and penetration depth advantage of the combination of SDT and PDT was demonstrated with the comparative treatment of *Candida albicans* biofilms and planktonic cells in vitro, with PDT, SDT and the combination of photodynamic and SDT, using chlorin e6 derivative photodithazine and Rose Bengal as the photosensitizers [153]. Interestingly, treatment of microbial cells with ultrasound waves can cause sonoporation, the reversible cracking of the cell membrane through which extracellular material can diffuse into the cells [154]. An additional advantage of SDT therefore is the enhancement of the intracellular uptake and retention of the aPDT photosensitizer through microbial ultrasonic sonoporation, which was originally exploited for gene transfection and targeted drug delivery [155,156]. In addition to the foregoing advantages, ultrasonic treatment was shown to weaken *Pseudomonas aeruginosa* biofilms by eroding the polysaccharide component of the extracellular polymetric substance matrix, enabling deeper biofilm penetration by the sensitizers [157]. The methodologies, pathogens and impacts of the three key studies on the combination of SDT and aPDT are summarized in Table 7.

## 12. Nanozyme Enhanced Antimicrobial Photodynamic Therapy

There has been escalating interest in research on nanozymes over the past decade and this has resulted in the birth of the new interdisciplinary area of nanozymology [158,159], which has found applications in many areas including antimicrobial and antibiofilm PDT [160]. Unlike natural enzymes, nanozymes are synthetic nanomaterials that possess enzyme mimetic properties such as peroxidase [161] and catalase [162] activities. Nanozyme-enhanced aPDT introduces the capability of augmenting PDT reactive oxygen species generation by incorporating nanozymes into the nanoconjugate system. The design, fabrication and application of nanoconjugates for nanozyme enhanced PDT in antimicrobial and antibiofilm research may be illustrated by the self-assembly of the cobalt (II)-5,10,15,20-tetrakis[4-(1,3,2-dioxaborinan-2-yl)phenyl]-21H,23H-porphyrin in the presence of 4,4′-bipyridine to form the nanozyme with surrogate catalase activity [163]. This nanozyme uses phenylboronic acid specific binding to bacteria and converts hydrogen peroxide, which is always over-expressed in the bacterial biofilm micro-environment, to form hydroxyl radicals, thus enhancing the photodynamic production of reactive oxygen species. The excellent inhibition of the formation of bacterial biofilm and bactericidal activity of the nanoconjugate may be attributed to the phenylboronic acid bacterial binding, which prevents the surface growth phases in the biofilm formation mechanism by close aggregation of the planktonic bacteria and photodynamic eradication as shown in Figure 5. The methodologies, pathogens and impacts of the three key studies on nanozyme-enhanced aPDT are summarized in Table 8. It can be noted from the data that all the studies reported in vitro and in vivo, indicating the novelty of the technology, and that a high degree of microbial eradication was reported, indicating its potential efficacy.

Organic-inorganic materials synthesized as crystalline porous nanoparticles known as metal-organic frameworks consist of a regular array of positively charged metal ions linked by surrounding organic molecules known as linkers. They assume porous cage-like structures with large pore sizes resulting in exceptionally large internal surface area [165]. Metal-organic-frameworks are emerging as formidable materials with a wide range of applications including direct applications as nanozymes [166]. An indirect application of metal-organic frameworks as nanozymes incorporating nanozyme activity-possessing molybdenum disulfide nanosheets was reported as part of a triple combination therapy involving photothermal, photodynamic and nanozyme peroxidase-like activity against ampicillin-resistant *Escherichia coli* and methicillin-resistant *Staphylococcus aureus* [164]. The UiO-66-NH-CO-MoS_2_ nanocomposite used in this study was constructed by conjugation of the amino-functionalized zirconium metal-organic-framework UiO-66 with the molybdenum disulfide carboxylic acid adduct using the amide bond formation reaction. The triple combination consisting of nanozyme conversion of hydrogen peroxide to form hydroxyl radicals, the photodynamic production of singlet oxygen and hydroxyl radicals, and photothermal temperature elevation, was augmented by the acceleration of glutathione oxidation, which further facilitated bacterial cell death as shown in Figure 6. Discovered at the University of Oslo, UiO-66 is a metal organic framework made up of [Zr_6_O_4_(OH)_4_] clusters linked with 1,4-benzodicarboxylic acid units [167].

Therefore, in addition to their well-known microbial cell eradication [168], nanozymes are reported to play the role of increasing reactive oxygen species to enhance PDT eradication of planktonic and biofilm-forming bacteria and fungi. While no clinical trials and clinical case studies have been reported so far, successes in pre-clinical studies with mice suggest that the nanozyme enhanced aPDT strategy may be well on its way to clinical applications [163,164].

## 13. General Discussion and Future Perspectives

Several research studies in this review illustrate the evolution of the applications of aPDT over its developmental trajectory and the improvement in the resultant inhibition of bacterial and biofilm growth, showing that there has been a technological revolution in this area over the past few decades. Due to diffusion and bacterial uptake limitations, many insoluble PDT photosensitizers showed poor disruption of the biofilms and inhibition of bacterial growth until the use of nanoconjugates as delivery agents. As science battled to solve these problems, amphiphilic sensitizers were developed; yet, the eradication of bacteria and biofilms and impact on disease remained low. Growing understanding of the contribution of biofilm formation and structure to bacterial resistance is helping to create new technologies and the discovery of new ways to overcome bacterial resistance. Besides the scientific excitement it caused, the recent nanotechnology revolution has made contributions to design and fabrication of new PDT approaches that have enhanced antimicrobial and antibiofilm strategies based on the new technology. Apart from facilitating targeted transport and delivery of photosensitizers, nanotechnology has enabled deep biofilm penetration-mediated destruction of the EPS matrix structure and closer access to the embedded pathogens. More importantly, it has enabled combination therapies involving PDT that successfully augment its bacterial and biofilm destruction mechanisms in many ways. Nanoconjugate-mediated combination with several known antibiotics such as vancomycin, minocycline, and cefepime have demonstrated the power of incorporating these drugs with the photodynamic photosensitizer in one nanoconjugate [107,108,109,110,111,112] while plasmonic nanoparticle-mediated combination with PTT [113,114,115,116,119,120,121,122,123] showed synergistic improvements in bacterial and biofilm eradication. Given that different antibiotic drugs have different mechanism and targets, the data presented by Pérez-Laguna et al. (2021) on the use of more than one antibiotic drug in combination with PDT suggests that more research is needed to evaluate synergism and additivity [21]. 

The low interest in studies of MHT combination applications against biofilm-forming microorganisms may be attributed to results showing that while both direct and alternating external magnetic fields achieve disruption of the biofilm matrix, more extensive biofilm matrix damage is obtained with direct magnetic field [135]. It could also be attributed to the fact that at present, besides the handheld devices, whole body magnetic applicators for humans across the world are limited to the patented MFH^®^300F, which is mainly used for inaccessible cancers such as glioblastoma and osteosarcoma [138]. When available in the open market, these devices are likely to be expensive. With these devices now coming into the market however, there is an important role for the combination of magnetic hyperthermia and PDT in the eradication of biofilm-forming microorganisms in the future. 

Other innovative new combinations include SDT and nanozyme enhanced aPDT. SDT is the combination of PDT and ultrasound in which the photosensitizer is energized by low-intensity ultrasound radiation, which penetrates deep enough to reach deep- seated bacterial infections [150,152]. Nanozyme enhanced aPDT exploits the incorporation of nanozymes into the photosensitizer-laden nanoconjugate. The nanozyme catalyzes the conversion of hydrogen peroxide, which is almost always present in high concentrations in the bacterial biofilm micro-environment, to hydroxyl radicals, adding to the concentration of reactive oxygen species that are produced by the photosensitizer-mediated production of singlet oxygen [163,164]. The ingenious approach of fabricating the nanozyme from the photosensitizer results in a common nanozyme and photosensitizer nanostructured material as shown in Figure 3. Once again, more pre-clinical research studies on nanozyme enhanced aPDT and SDT will benefit in the war against biofilm-forming microorganisms in the future. As predicted by Wang et al. (2017), the great promise of SDT is the treatment of topically inaccessible microbial infections [169].

Recently debuting is the novel combination of aPDT with CAP, which has been successfully used on its own, particularly in periodontal disease and burn wounds, which are notorious for biofilm-mediated bacterial resistance. The investigation of the combination of CAP and aPDT appears to have been triggered by a series of comparative studies of the two technologies that revealed the possibility of their synergistic combination [75,141,142,143,144]. Indeed, the scientific community is anticipating the outcomes of a bold new combination of aPDT with CAP which is still in the doctoral thesis stage [75]. More of these kinds of studies will have to be conducted in the future in order to validate the combination therapy consisting of CAP and aPDT. 

## 14. Challenges and Limitations and How They Can Be Overcome

A number of challenges still mitigate the clinical excellence of aPDT and combinations with non-invasive antimicrobial technologies that have been discussed in this review. For example, biofilm diffusion and bacterial uptake of the photosensitizer is still a limitation of aPDT [14]. As a result, much research is dedicated to development of novel photosensitizers with improved biofilm diffusion and bacterial uptake. Even with the demonstrated efficacy improvement, the persistent light penetration depth limitations of aPDT combinations may be overcome by the advancing research in SDT [152]. Furthermore, the bacterial uptake of aPDT photosensitizers can be improved by bacterial sonoporation in combination with SDT. The combinations with photothermal therapy are also limited by light penetration depth to skin wound therapeutic applications [126]. The low interest in combinations with MHT may be attributed to the low biofilm weakening efficacy of magnetic hyperthermia [136] and the high cost of the applicator [138]. Although a number of devices are now commercially available for the combination with CAP, the combination is still limited to skin and low depth wound infections [34,35,36,37]. This suggests that more research is required on indirect CAP, which uses plasma activated fluid [38]. While nanozyme enhanced combinations show a remarkable efficacy enhancement, it must be noted that, like aPDT, they are equally plagued by limitations in light penetration depth. 

## 15. Conclusions

The remarkable development of aPDT has had an impact in the current war against biofilm-induced bacterial resistance. Pang et al. (2020) notes that the rate of scientific innovation to combat microbial infection is outpaced by the rate of evolutionary development of microbial resistance [147]. However, although some mechanisms for bacterial resistance against it have been described, the rapid technological revolution in this area may be outpacing the genetic and environmental mechanisms for the development of bacterial resistance [16]. Nanotechnology-mediated aPDT and its associated innovations have clearly made many significant contributions in the war against biofilm-induced bacterial resistance. This review has described aPDT and six non-invasive combination technologies, highlighting their strengths and weaknesses. The intention is to stimulate research aimed at transitioning the strong aPDT combinations to clinical applications, ameliorating weaknesses, and exploring other combinations. 

While aPDT alone continues to make contributions to clinical practice, some of the combinations are more likely to remain in exploration for quite some time. For example, aPDT in combination with MHT will have to continue in experimental and pre-clinical studies until biofilm eradication by MHT is proven and ubiquitous availability of the MHT applicators is realized. While the combination of aPDT with antibiotic chemotherapy is likely to continue to flourish, reaching clinical milestones with previously unexplored antibiotic drugs, it will still be limited to skin and shallow wound therapy. Similarly, despite the reported in vitro synergism and efficacy of in vivo studies, the combination of PTT and aPDT is more likely to be limited to topical and shallow infections, as will CAP in combination with aPDT. However, the advancement of the SDT combination with aPDT will allow for treatment of deeper-lying and obscure disease. Nanozyme enhanced aPDT is still at its earliest phases in the proof of concept and has a long way to go and similar hurdles to overcome if it is to make a significant contribution beyond topical and shallow infections. aPDT and some non-invasive combinations are also making contributions to various dental therapeutics and are likely to continue this trend.

## Figures and Tables

**Figure 1 ijms-23-03209-f001:**
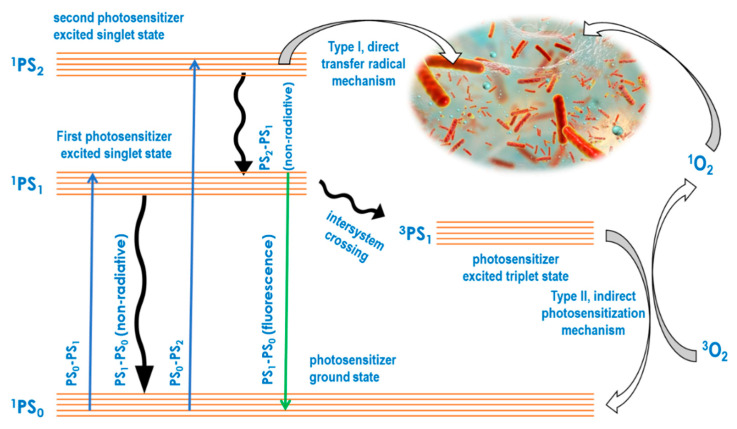
Jablonski diagram to illustrate the aPDT type I and II mechanisms.

**Figure 2 ijms-23-03209-f002:**
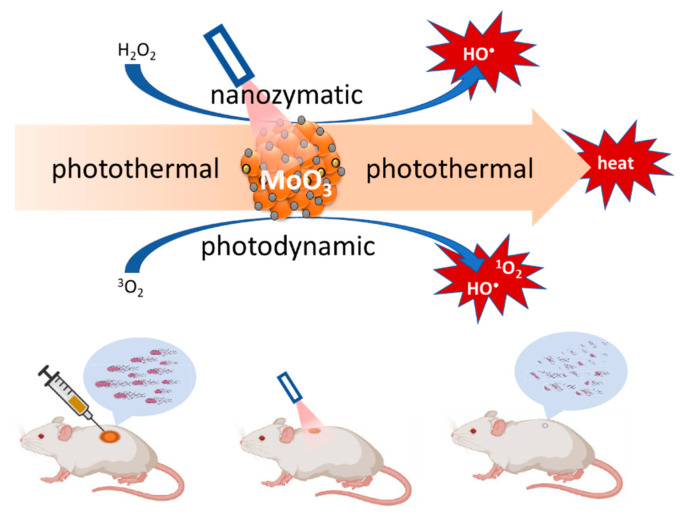
Illustration of the triple therapy combination of PTT, PDT, and nanozyme effect of molybdenum trioxide nanoparticles.

**Figure 3 ijms-23-03209-f003:**
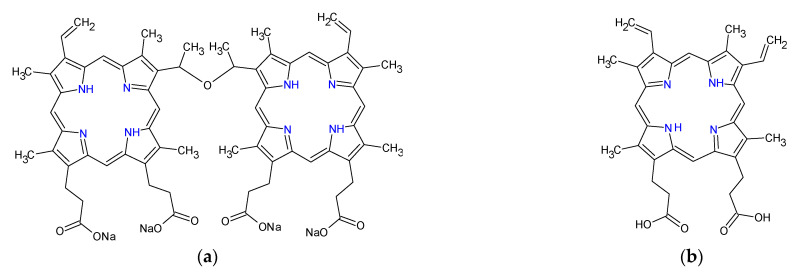
Chemical structures of sinoporphyrin sodium (**a**) and protoporphyrin IX (**b**).

**Figure 4 ijms-23-03209-f004:**
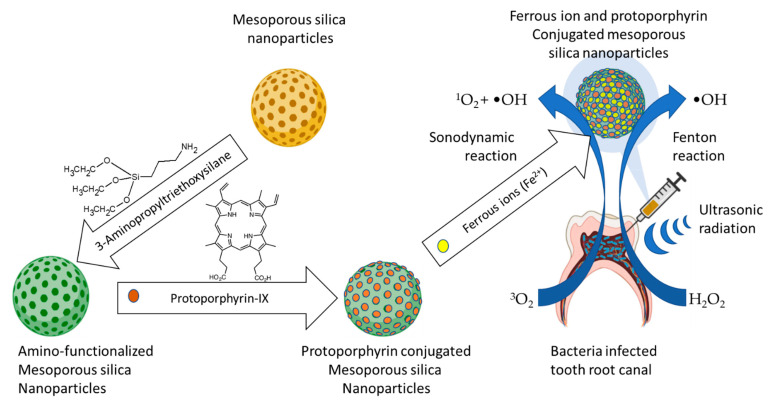
Illustration of the Fenton reaction-enhanced SDT, a combination of SDT with the Fenton reaction generation of hydroxyl radicals (Guo et al., 2021) [152].

**Figure 5 ijms-23-03209-f005:**
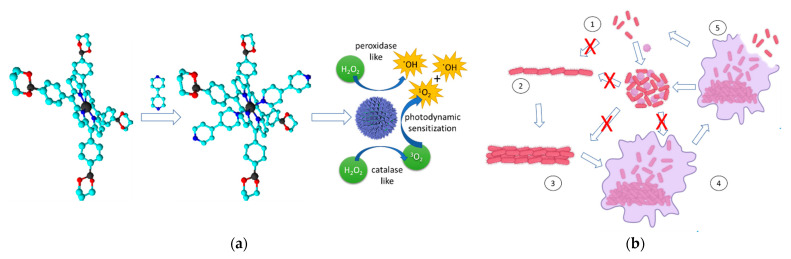
The self-assembly nanozyme formation mechanism and the planktonic bacterial cell aggregation mechanism leading to photodynamic bacterial cell death and biofilm structure destruction. (**a**) Bipyridine mediated self-assembly of the porphyrins to form the porphyrin nanozyme with peroxidase/catalase mimic activities. (**b**) Planktonic cell aggregation mechanism reported by Hu et al. (2022) [163]. (1) planktonic microbial form, (2) surface adhesion, (3) colony formation and maturation, (4) biofilm formation, (5) microbial detachment from biofilm.

**Figure 6 ijms-23-03209-f006:**
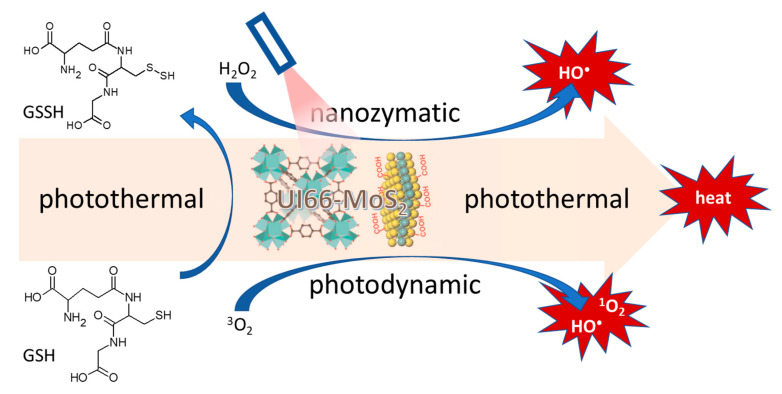
Nanozyme, photothermal, photodynamic, and glutathione oxidation activity of molybdenum disulfide sheet conjugated UIO-66 metal-organic-framework [167].

**Table 1 ijms-23-03209-t001:** Antibacterial photodynamic therapy alone.

Photosensitizer	Nanoconjugate System Used	Gram Negative	Gram Positive	Study Phase	Impact Target	Ref
polyethylenimine-chlorin(e6) and tris-cationic-buckminsterfullerene	dendrimer nanoconjugate	*E. coli* *P. mirabilis* *P. aeruginosa*	*S. aureus* *E. fecalis*	in vitro in vivo	biofilm + planktonic	[57]
1-oxo-1H-phenalen-2-yl methyl pyridinium chloride (SAPYR) and 1-oxo-1H-phenalen- 2-yl-methyl-dodecan-1-aminium chloride (SA-PN-05)	None	*E. coli*	*A. naeslundii* *S. mutans*	in vitro	biofilm + planktonic	[58]
porfimer sodium (hematoporphyrin derivative)	None	*F. nucleatuma*		in vitro	biofilm	[61]
Methylene blue	None	*E. coli*, *K. pneumoniae*, *P. aeruginosa*, *S. marcescens*, *H. influenzae*	*C. albicans*, *E. faecalis*, *S. aureus*, *S. pneumoniae*	in vitro in vivo	biofilm	[61]
Chlorin-e6	None	*H. influenzae*	*M. catarrhalis*, *S. pneumoniae*	in vitro	biofilm + planktonic	[63]
Protoporphyrin IX and Methylene blue	None	*A. baumannii*		in vitro	biofilm + planktonic	[64]

**Table 2 ijms-23-03209-t002:** Antibacterial photodynamic therapy in combination with antibiotic therapy.

Photosensitizer and Nanoconjugate System Used	Antibiotic Drug Used	Gram Negative	Gram Positive	Study Phase	Impact/Target	Ref
indocyanine green and ethylenediamine tetraacetate	vancomycin minocycline for MRSA, amikacin and cefepime MRPA.	*P. aeruginosa*	*S. aureus*	in vitro	biofilm + planktonic synergy	[107]
amoxicillin-coated nanoparticles of gold	amoxicillin	*P. aeruginosa*	*S. aureus*	in vitro	biofilm + planktonic	[108]
zeolitic imidazolate framework-8-polyacrylic acid loaded with methylbenzene blue (ZIF-8-PAA-MB@AgNPs@Van-PEG)	vancomycin	*E. coli*	*S. aureus*	in vitro in vivo	biofilm + planktonic/synergy	[109]
ALA-PDT therapy	clarithromycin, moxifloxacin, rifampicin, ethambutol hydrochloride, and levofloxacin	*M. fortuitum*	*M. abscessus*, *M. gordonae*, *M. gilvum*,	clinical case study	biofilm + planktonic (wound healing)	[87]
ALA-PDT therapy	amikacinand rifampicin and clarithromycin	*M. fortuitum*		clinical case study	biofilm + planktonic (wound healing)	[88]
protoporphyrin IX	ceftriaxone	*E. coli*, *P. aeruginosa*	*S. aureus*	in vivo	biofilm + planktonic (wound healing)	[110]
indocyanine green	amoxicillin		*S. milleri*	clinical case study	100% healing	[111]

**Table 3 ijms-23-03209-t003:** Antibacterial photodynamic therapy in combination with photothermal hyperthermia therapy.

Photosensitizer and Nanoconjugate System Used	Photothermal Therapy Agent Used	Gram Negative	Gram Positive	Study Phase	Impact/Target	Ref
Toluidine blue and withindocyanine green	indocyanine green		*S. mutans*	in vitro	biofilm + planktonic/enhanced efficacy	[73]
indocyanine green loaded SPIONs	superparamagnetic iron oxide nanoparticles	*E. coli*, *K. pneumoniae*, *P. aeruginosa*,	*S. epidermis*	in vitro	biofilm + planktonic/synergistic	[26]
indocyanine green loaded mesoporous nanoparticles	mesoporous polydopamine nanoparticles	*E. coli*, *K. pneumoniae*, *P. aeruginosa*	*S. aureus*	in vitro	biofilm + planktonic/synergistic	[114]
toluidine blue coated gold nanorods	gold nanorods		MRSA	in vitro	biofilm + planktonic/synergistic	[117]
black phosphorus nanosheets conjugated gold nanoparticles BP@AuNP	BP@AuNP	*E. coli*	*S. aureus*	in vitro in vivo	biofilm + planktonic/synergistic	[118]

**Table 4 ijms-23-03209-t004:** Antibacterial photodynamic, nanozyme, and photothermal hyperthermia tritherapy combinations.

Photodynamic, Nanozyme, and Photothermal Hyperthermia Tritherapy Agent Used	Gram Negative	Gram Positive	Study Phase	Impact/Target	Ref
molybdenum trioxide nanodots	*E. coli*	MRSA	in vitro in vivo	biofilm + planktonic/10 mm wound closure in 7 days	[119]
Ag-nanoparticle decorated MoS_2_@polydopamine nanosheets		MRSA	in vitro in vivo	biofilm + planktonic/10 mm wound closure in 7 days	[122]
Antibacterial photodynamic, nanozyme, and photothermal hyperthermia tritherapy combinations					
polydopamine (PDA)- IR820-Daptomycin		*S. aureus*	in vivo	biofilm + planktonic/inhibition on titanium implants	[123]

**Table 5 ijms-23-03209-t005:** Comparative studies of antibacterial photodynamic therapy and magnetic hyperthermia therapy.

Photosensitizer and Nanoconjugate System Used	Photothermal Therapy Agent Used	Gram Negative	Gram Positive	Study Phase	Impact/Target	Ref
curcumin superparamagnetic iron oxide nanoconjugate	superparamagnetic iron oxide		*S. aureus*	in vivo	planktonic/complete eradication	[74]
Magnetic targeting studies and antibacterial photodynamic therapy						
toluidine-blue ortho, nanoemulsion encapsulated superparamagnetic iron oxide	nanoemulsion encapsulated superparamagnetic iron oxide		*S. mutans*	in vitro in vivo	targeting, imaging	[136]

**Table 6 ijms-23-03209-t006:** Antibacterial photodynamic therapy in combination with cold atmospheric pressure plasma therapy.

Photosensitizer and Nanoconjugate System Used	Cold Atmospheric Pressure Plasma	Gram Negative	Gram Positive	Study Phase	Impact/Target	Ref
indocyanine green direct treatment without nanoconjugate	home made device, 20 kHz/30 kV		MRSA	in vitro	biofilm + planktonic logCFU/mL reduction: 3.52, CAPP: 3.61	[76]
methylene blue direct treatment without nanoconjugate	Plasma Pen™, He (98%) + O_2_ (2%) 6 bar and 1 kV		*E. faecalis*	in vitro	biofilm, AH Plus push-out bond strength: aPDT: 2.44, CAPP: 3.54	[141]
HELBO^®^ Blue Photosensitizer	plasma jet (CAP1), dielectric barrier discharge (CAP2)		*E. faecalis*	in vitro	planktonic, logCFU/mL reduction: aPDT: 5.25, CAP1: 5.4 CAP2: 5.8	[142]
HELBO^®^ Blue Photosensitizer	Plasma ONE device (420–1220 Hz, 7.2 V)	*A. baumannii*	*S. aureus*	in vivo	planktonic, biofilm, aPDT: complete eradication, CAP: infection depth dependant	[143]
toluidine blue direct treatment without nanoconjugate	dielectric barrier discharge (25-kHz, 5-kV, He + 0.5% O_2_)		*E. faecalis*	in vitro	planktonic, logCFU/mL reduction: aPDT: 2.156, CAP: 0.17.	[144]

**Table 7 ijms-23-03209-t007:** Combination of antibacterial photodynamic therapy with sonodynamic therapy.

Photosensitizer and Nanoconjugate System Used	Sonodynamic Therapy	Gram Positive	Study Phase	Impact/Target	Ref
uroporphyrin and coproporphyrin III	home-made light source/ultrasound generator	*S. aureus*	in vitro	planktonic, biofilm, no difference between aPDT and SDT	[150]
Fe^2+^ and protoporphyrin IX conjugated mesoporous silica nanoparticles	home-made light source/ultrasound generator	*E. faecalis*	in vitro	planktonic, biofilm, no difference between aPDT and SDT	[152]
chlorin e6 derivative Photodithazine^®^ rose bengal	Sonidel SP100 sonoporator (sonar 1 MHz and pulse repetition frequency of 100 Hz	*C. albicans*	in vitro	planktonic, biofilm, logCFU/mL reduction: aPDT/SDT: 2.08/3.39, PDT/SDT: eradication	[153]

**Table 8 ijms-23-03209-t008:** Nanozyme enhanced antimicrobial photodynamic therapy.

Nanozyme Nanoconjugate System Used	Photodynamic Reaction	Gram Negative	Gram Positive	Study Phase	Impact/Target	Ref
silver nanoparticle decorated molybdenum disulphide nanosheet-capped iron oxide nanozyme	nanozyme peroxidase-like production of reactive oxygen species	*E. coli*	*S. aureus*, *B. subtilis*, *MRSA*, and *C. albicans*,	in vitro	planktonic, concentration dependant eradication	[160]
cobalt-5,10,15,20-tetrakis[4-(1,3,2-dioxaborinan-2-yl) phenyl]-21H,23H-porphyrin 1,4-bipyridyl self assembled nanozyme	nanozyme catalase-like and peroxidase-like production of reactive oxygen species	*E. coli* *P. aeruginosa*	*S. aureus* *B. amyloliquefaciens*	in vitro in vivo	planktonic/biofilm, >95% bacterial count reduction	[163]
molybdenum disulphide nanosheet-amide bond conjugated metal-organic-framework	nanozyme catalase-like and peroxidase-like production of reactive oxygen species	*E. coli*	MRSA	in vitro in vivo	planktonic/biofilm, >99.7% bacterial count reduction	[164]

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
