# Peer review of "Applications of Antimicrobial Photodynamic Therapy against Bacterial Biofilms"

_ijms, 2022, doi:10.3390/ijms23063209_

Round 1

Reviewer 1 Report

The review presented by Songca and collaborator needs some improvements before its possible publication. Some comments are reported below:

-A figure summarizing the introduction is highly recommend.

-Authors need to introduce some tables summarizing the several approach presented with the main findings.

-A limitation of the approach should be introduced in the manuscript, and how in the future limitations can be overcome

-In the conclusion section, authors should remark the main theme treated in the review and how this work can help scientists in the field.

-English grammar should be rigorously checked.

Author Response

Reviewer 1

The review presented by Songca and collaborator needs some improvements before its possible publication. Some comments are reported below:

Reviewer

-A figure summarizing the introduction is highly recommend.

Response

An excellent recommendation, thanks. This is done on page 1.

Reviewer

-Authors need to introduce some tables summarizing the several approach presented with the main findings.

Response

This is yet another excellent recommendation. In this regard we have introduced 8 tables to summarise the approaches discussed in the manuscript listing the nanotechnologies, the stages in the developmental trajectory (in-vitro, in-vivo, clinical experiments and applications), while mapping the effectiveness and references of the examples used. These are found on pages 15-19.

Reviewer

-A limitation of the approach should be introduced in the manuscript, and how in the future limitations can be overcome

Response

This yet another great improvement to the manuscript. It is done by introducing a short section 14. Challenges and limitations and how they can be overcome.

Reviewer

-In the conclusion section, authors should remark the main theme treated in the review and how this work can help scientists in the field.

Response

This is also considered to be a great improvement to the manuscript. The following lines have been introduced.

This review has described aPDT and six non-invasive combination technologies, highlighting their strengths and weaknesses. The intention is to stimulate research aimed at taking the strong aPDT combinations to clinical applications, ameliorating the weaknesses, and exploring other combinations.

Reviewer

-English grammar should be rigorously checked.

Response

Grammarly, set to the American English, was run twice.

Reviewer 2 Report

Songca and Adjei reviewed the recent development of Antimicrobial photodynamic therapy in fighting bacterial biofilm. This work has good logic and systematicness. In my opinion, this manucript is suitable for publication in IJMS after addression the following problems.

  1.  In line 273, the figure 1 should be figure 2. Besides, there is two pictures of figure 2.
  2.  The subtitle is recommended to revise as "Combination with cold atmospheric pressure plasma therapy"
  3.  In the section 11,  the author introduced the advantage of  Sonodynamic Therapy,  which make this section discordant. it is better to replace with "Combination with Sonodynamic therapy", For example, the article"Antimicrobial sonodynamic and photodynamic therapies against Candida albicans" is a good example, ect.
  4.  Some references which have direct relation with manuscript should discussed and cited. for example, J. Mater. Chem. B, 2020, 10700-10711; View2(1), 20200065.
  5.  Future directions and challenges of antimicrobial photodynamic therapy also should be discussed.

Author Response

Reviewer 2

Songca and Adjei reviewed the recent development of Antimicrobial photodynamic therapy in fighting bacterial biofilm. This work has good logic and systematicness. In my opinion, this manuscript is suitable for publication in IJMS after addressing the following problems.

Reviewer

In line 273, the figure 1 should be figure 2. Besides, there is two pictures of figure 2.

Response

Thanks to the observation of the reviewer, all figures have been correctly labelled and serially numbered. Citations of the figures in the text have also been corrected to reference the correct diagrams.

Reviewer

The subtitle is recommended to revise as "Combination with cold atmospheric pressure plasma therapy"

Response

Thanks for this. Section 10 has been changed accordingly and now reads as follows:

10. Combination with cold atmospheric pressure plasma therapy

Reviewer

In the section 11, the author introduced the advantage of Sonodynamic Therapy,  which make this section discordant. it is better to replace with "Combination with Sonodynamic therapy", For example, the article"Antimicrobial sonodynamic and photodynamic therapies against Candida albicans" is a good example, ect.

Response

Once again the reviewer has pointed out a beneficial correction. Section 11 now reads as follows:

11. Combination with Sonodynamic therapy

Reviewer

Some references which have direct relation with manuscript should discussed and cited. for example, J. Mater. Chem. B, 2020, 10700-10711; View, 2(1), 20200065.  Future directions and challenges of antimicrobial photodynamic therapy also should be discussed.

Response

Indeed, the responsiveness of nanoconjugates used in antibacterial photodynamic therapy to pH, [O2], [H2O2], and enzymes has been exploited for triggering nanomaterial-based photosensitizer and chemotherapy drug targeting, delivery and release. Without delving much into the content of the two recommended reviews, their important contribution to the current manuscript have been captured with the following sentence and referenced accordingly.

Responsiveness to the biofilm microenvironmental characteristics such as acidity, hypoxia, enzyme and hydrogen peroxide concentration, has been exploited for triggering nanomaterial-based photosensitizer and chemo-therapy drug delivery and release, and for enhancing disease site and cell targeting.

Reviewer 3 Report

The review concerns the use of photodynamic therapy, alone and as part of more complex protocols, for eradication of biofilms. The films appear to be resistant to treatment with antibiotics during the conventional clinical approach to infections. While this report does summarize much of the current research effort, many of the terms used are not defined and some extraneous material is included. How the techniques suggested might be applied to treatment of such films in vivo is not well explained. 

Looking sequentially through the report, what is a ‘strong filter effect’ (line 36)? What is ‘cold atmospheric pressure plasma therapy”? References 29 and 30 are not peer-reviewed and consist of a dissertation report and a conference abstract. Ref. 111 refers to a PLoS One report that describes the technique using agar plates, electrodes, treatment with 3.5 kV at a frequency of 4 Hz. This is reported to generate ozone, nitrogen dioxide and other reactive species. Exactly how this could be used in a clinical setting is not revealed. In Ref. 111, it is reported that the overall effect is similar to the efficacy of 0.2% chlorhexidine which might be easier to apply.  Material on lines 354-373 appears to be mainly speculation. 

PDT can not overcome hypoxia (line 50). An exogenous oxygen supply might be helpful. Since there are no protocols suggested for the management of biofilms involving ionizing radiation or immunotherapy, why are these mentioned (lines 52-53). 

There is little critical discussion of the assorted modalities involving PDT (lines 149-155). Several approaches are proposed, but are they effective? In the discussion of combinations involving hyperthermia, are these in vitro or in vivo studies? Eradication by PDT and associated approaches in petri dishes is usually not a challenge. This also applies to material in section 11 (sonodynamic therapy). Are the ‘promising results’ (line 379) in vitro or in vivo? 

In the References, the style is varied. Sometimes every word in a citation is capitalized, e.g., Ref 5,  sometimes not (Ref 6).      

Minor points: check spelling of ‘bacterial’ (line 127); ‘triple’ (line 273). 

Author Response

Reviewer 3

Reviewer

The review concerns the use of photodynamic therapy, alone and as part of more complex protocols, for eradication of biofilms. The films appear to be resistant to treatment with antibiotics during the conventional clinical approach to infections. While this report does summarize much of the current research effort, many of the terms used are not defined and some extraneous material is included. How the techniques suggested might be applied to treatment of such films in vivo is not well explained.

Response

Undefined uncommon terms were identified including magnetic hyperthermia therapy, cold atmospheric pressure plasma therapy, photothermal hyperthermia therapy, extracellular polymeric substance matrix, nanozyme, nanozyme enhanced photodynamic therapy, sonodynamic therapy, metal-organic-frameworks, and UI66.

The manuscript has been revised and definitions for each of these terms are now found across the manuscript in the sections where each is presented, e.g. magnetic hyperthermia therapy (lines 60-64), cold atmospheric pressure plasma therapy (lines 72-76), photothermal hyperthermia therapy (lines 362-365), extracellular polymeric substance matrix (lines 102-107), nanozyme (lines 611-613), nanozyme enhanced photodynamic therapy (section 12, lines 606-656), sonodynamic therapy (lines 703-707), metal-organic-frameworks (lines 631-635), and UI66 (lines 653-656).

Reviewer

Looking sequentially through the report, what is a ‘strong filter effect’ (line 36)? What is ‘cold atmospheric pressure plasma therapy”? References 29 and 30 are not peer-reviewed and consist of a dissertation report and a conference abstract.

Response

The reviewer has spotted a term that is not defined and one that is inappropriately placed in this manuscript. For this reason, the term has been removed. Instead, examples of mechanisms of resistane are more clearly listed, including ABCG2 mediated efflux, DNA damage repair, procaspase damage in-duced inhibition of apoptosis, heat shock protein upregulation, hypoxia, and antioxidant defense mechanisms.

The references [75] and [76] relate to publications that are not peer-reviewed. Reference [75] refers to a doctoral thesis still in the institutional library at the University of Greifswald. Reference [76] refers to a paper published as part of the proceedings of the Medical Technologies Congress in 2020. The point being made here is that the combination of cold atmospheric pressure plasma is so novel that research studies on this combination have not yet even been peer-reviewed. I therefore request leave to use these references to make this point.

72  Aly, F. and Daeschlein G. Synergistic antimicrobial effects of cold atmospheric pressure plasma and photodynamic therapy against common skin and wound pathogens in-vitro, Universität Greifswald 2021, Universitätsmedizin, Faculty of Medicine, Polyclinic for Skin Diseases

73  Sirek, B.; Fidan, D.; Özdemir, G.D.; Bakay, E.; Ercan U. K. and TopaloÄŸlu, N. Comparison of the Antibacterial Effects of Photodynamic Therapy and Cold Atmospheric Plasma on Methicillin-Resistant Staphylococcus aureus, Medical Technologies Congress (TIPTEKNO) 2020, 1-4. https://doi.org/10.1109/TIPTEKNO50054.2020.9299269

Reviewer

Ref. 142 refers to a PLosONE report that describes the technique using agar plates, electrodes, treatment with 3.5 kV at a frequency of 4 Hz. This is reported to generate ozone, nitrogen dioxide and other reactive species. Exactly how this could be used in a clinical setting is not revealed. In Ref. 111, it is reported that the overall effect is similar to the efficacy of 0.2% chlorhexidine which might be easier to apply.  Material on lines 354-373 appears to be mainly speculation.

Response

In asking how exactly how this could be used in a clinical setting the reviewer points out a reasonable expectation from readers. In addition to the identification of devices now in clinical cold atmospheric pressure plasma use, this has been addressed by including the recommended description in lines 524-531, as follows:

The combination of cold atmospheric pressure plasma therapy with antimicrobial photodynamic therapy in clinical treatment of septic wounds for example typically involves one of the cold atmospheric pressure plasma devices such as the dielectric barrier discharge to generate the plasma jet. The plasma jet produces sufficient light for the antimicrobial photodynamic therapy production of reactive oxygen species, including singlet oxygen and hydroxyl radicals.

The reviewer is correct. The material on lines 354-373 (now 429-444) appears to be mainly speculation. The reason is once again to make the point that the combination of cold atmospheric pressure plasma is so virgin that research studies on this combination have not yet even been peer-reviewed. I therefore request leave to speculate to make this point.

Reviewer

PDT cannot overcome hypoxia (line 50). An exogenous oxygen supply might be helpful. Since there are no protocols suggested for the management of biofilms involving ionizing radiation or immunotherapy, why are these mentioned (lines 52-53).

Response

The reviewer is correct, PDT cannot overcome hypoxia. Only when combined with techniques such as cold atmospheric pressure plasma therapy and nanozyme enhanced photodynamic therapy, does PDT have the potential to overcome hypoxia. The reason for this is that cold atmospheric pressure plasma therapy introduces an exogenous oxygen supply of reactive oxygen species from the plasma jet (in the case of the direct cold atmospheric pressure plasma therapy) or from the plasma activated fluid (in the case of the indirect cold atmospheric pressure plasma therapy). With nanozyme enhance photodynamic therapy, the hydrogen peroxide which is overexpressed in the microbial microenvironment is converted to oxygen and reactive oxygen species, thus diminishing the hypoxia effect.

Thanks for question concerning the mention of combination of antimicrobial photodynamic therapy with radiation and immunotherapy. Although protocols involving these combinations are not included in this review, they are mentioned to indicate the width of the range of combinations with antimicrobial photodynamic therapy that have been investigated.

Reviewer

There is little critical discussion of the assorted modalities involving PDT (lines 149-155). Several approaches are proposed, but are they effective?

Response

The reviewer has pointed out an important omission of the critical discussion of each of the assorted modalities. This has been addressed across the manuscript by introduction of critical discussions in the discussion of each modality at the end of the section where it is presented. For example lines 286-294, 470-486, 530-536,

Reviewer

In the discussion of combinations involving hyperthermia, are these in vitro or in vivo studies? Eradication by PDT and associated approaches in petri dishes is usually not a challenge. This also applies to material in section 11 (sonodynamic therapy). Are the ‘promising results’ (line 379) in vitro or in vivo?

Response

An important clarity-seeking question is raised by the reviewer here. The first paragraph in section 5 has been revised to address the question. It now reads as follows (lines 169-177):

PDT has been used in combination with several non-invasive therapeutic approach-es with additive and synergistic efficacy enhancement outcomes in most studies con-ducted in-vitro, as well as in preclinical and clinical applications. In this regard, studies and applications have been reported for aPDT in combination with antibiotic chemotherapy PTT, MHT, CAP, endodontic debridement. Stud-ies and applications have also been reported in combinations using multiple PDT photosensitizers, in what could be termed multiple photosensitizer combination aPDT. In many multiple photosensitizer combination studies, inorganic-organic and organic-inorganic hybrid photosensitizers are recognized.

Similarly, in section 11, the first paragraph has been revised, and now reads as follows (lines 558-572):

SDT is an innovative combination of PDT with ultrasound in that it involves expos-ing diseased tissues to chemical compounds which produce reactive oxygen species up-on sensitization by means of low-intensity ultrasound. In addition to the reported anti-cancer research applications, it has been applied to antimicrobial research studies with promising results in-vitro. The evaluation of SDT may be illustrated with the comparison between PDT and SDT on Staphylococcus aureus biofilm samples. This research compared ultrasound treatment, photodynamic treatment and the combined ultrasound and photodynamic treatment, and found that the combined treat-ment reduced the planktonic and biofilm of Staphylococcus aureus more than ultrasound treatment alone, and more than photodynamic treatment alone. As correctly pointed out by Fan et al (2021), after much mechanistic and efficacy studies in-vivo clinical SDT is imminent.

Reviewer

In the References, the style is varied. Sometimes every word in a citation is capitalized, e.g., Ref 5, sometimes not (Ref 6).     

Response

All references have now been revised and are presented in MDPI style.

Reviewer

Minor points: check spelling of ‘bacterial’ (line 127); ‘triple’ (line 273).

Response

The entire document was again spelling checked using South African English Microsoft Word settings.

Round 2

Reviewer 1 Report

Although the manuscript following my indication reported the required improvement, authors should perform some work before acceptance. Table should be appropriately reported within the text and no one after another without any sense. Table 6 was not cited in the text and the term table should be written with the first word as capital letter.

Round 2

Reviewer

Although the manuscript following my indication reported the required improvement, authors should perform some work before acceptance.

Response

The reviewer points out a very important improvement for the readability of the review and for the tables to make sense. This has been dome. Each table is now inserted in the correct place next to the relevant discussion. Thanks.

Reviewer

Table should be appropriately reported within the text and no one after another without any sense. Table 6 was not cited in the text and the term table should be written with the first word as capital letter. My apologies. It is cited in line 429 now.

Response

Once again, a valuable observation. This recommendation has been effected (1) each table is referred to in the relevant text, and (2) the word table in the text has been written as Table. Thank you. 

Reviewer 2 Report

After carefully revised, the manuscript can be received for publication in IJMS in my opinion.

Round 2

Reviewer

After carefully revised, the manuscript can be received for publication in IJMS in my opinion.

Response

Thank you. Further revisions have been made.

Reviewer 3 Report

This report relates to the potential role of photodynamic processes in dealing with microbial infections. One of the authors of this report is cited only twice, but one can compose a review article without necessarily being involved in the research effort. 

It was noted before that many abbreviations are missing. Some are now provided in the Abstract which is helpful although these are perhaps better provided in the text when the various terms are first used. Some of the procedures appear to be poorly adapted to clinical practice, e.g., MHT. This involves temperature elevation by magnetic nanoparticles. CAP is said to involve creation of plasmas created by high-voltage electrodes. This does appear to be used in clinical practice. The relevance of some studies to clinical treatment is, however, unclear. For example, Ref. 74 is cited (line 173) as an example of MHT efficacy but this is an in vitro study as are many of the other references cited, e.g., 146. 

Few of the references cited in the Tables relate to in vivo studies. Other statements are not entirely pertinent, e.g., the notation that a significant death rate from microbial infections is anticipated (line 163). This may be true, but the impact of antimicrobial PDT will likely be minimal for treatment of systemic disease. 

The major publications are covered in this review. Wheether the procedures described will have widespread utilization is unknown since many of the techniques are not generally available and in vivo efficacy is infrequently established. So while an updated review of this material can be useful, so would comments on the likelihood that techniques proposed will reach general clinical practice or whether they will be confined to a few centers.

Round 2

Reviewer

This report relates to the potential role of photodynamic processes in dealing with microbial infections. One of the authors of this report is cited only twice, but one can compose a review article without necessarily being involved in the research effort.

Response

Thanks. This observation is accurate. We have a culture of publishing with our doctoral students and postdoctoral fellows. They grow in the process. Dr. Adjei was my doctoral student. We have published 5 papers together. His citations will increase in time.

Reviewer

It was noted before that many abbreviations are missing. Some are now provided in the Abstract which is helpful although these are perhaps better provided in the text when the various terms are first used.

Response

Thanks. All abbreviations are now given at first mention after the abstract. The abstract now reads much more clearly. Thanks.

Reviewer

Some of the procedures appear to be poorly adapted to clinical practice, e.g., MHT. This involves temperature elevation by magnetic nanoparticles.

Response

Thanks for this observation, which is correct. For example MHT is only just starting to debut into the clinic for applications against hard to reach cancers, e.g. brain, liver and pancreatic. The single applicator in the world is in Germany and costly. MHT for antimicrobial applications is still at infancy. It is mentioned in this review to stimulate interest to find ways in which it can be adopted for antimicrobial applications.

Reviewer

CAP is said to involve creation of plasmas created by high-voltage electrodes. This does appear to be used in clinical practice.

Response

Thanks. On the contrary, CAP alone in the clinic is ubiquitous, practiced by clinicians as plasma medicine. In the review we have mentioned this in the paragraph from line 52-71, including the clinical devices used. However, CAP in combination with aPDT has not been reported in the clinic. In the review we have mentioned this in section 10 in the lines 413-431, references 34-37. What has appeared in the literature is the comparison of CAP and aPDT in-vitro and in-vivo. Once again the point being made here is that a technology that appears to hold great promise is currently being studied from various angles, with a doctoral study as the first experimental investigation of the combination.

Reviewer

The relevance of some studies to clinical treatment is, however, unclear. For example, Ref. 74 is cited (line 173) as an example of MHT efficacy but this is an in vitro study as are many of the other references cited, e.g., 146.

Response

Thank you for pointing this out. In this review we have adopted an approach of tracing each of the technologies discussed all the way to basic studies and proofs of efficacy in-vitro, working our way up to clinical applications. The reference 74 is an interesting case in point because it reports the synthesis and in-vitro studies of a nanoconjugate that could be investigated for the combination of MHT and aPDT. Yet the experiment was never done. It is limited to comparative studies! Its relevance is in the synthesis of the nanoconjugate and the comparative studies of MHT and aPDT. This is equally true for the reference 146, which reports the comparative studies of CAP with aPDT. The purpose of the review is not to present clinical applications only, because indeed some of the technologies have not reached clinical applications. The purpose is exactly to point out the current position of each of the technologies along the value chain (synthesis, in-vitro, in-vivo, clinical trials, clinical case studies).

Reviewer

Few of the references cited in the Tables relate to in vivo studies.

Response

Thanks for the observation, which is true. There are only 12 in vivo studies compared to 27 in-vitro studies. Taken collectively, the data suggests that most combination therapy studies with aPDT are still located at in-vitro, although a few in-vivo and clinical applications are reported. The review has dedicated a large portion of the discussion to the introduction of the combinations and illustration of the proof of in-vitro efficacy.

Reviewer

Other statements are not entirely pertinent, e.g., the notation that a significant death rate from microbial infections is anticipated (line 163).

Response

Thanks for the observation. At first glance the global death rate crisis predicted by the WHO appears indeed to lack pertinence. However, the point being made in the paragraph is that to avert the WHO prediction, which is based on the current death trends due to declining effectiveness of traditional antibiotics, every technology that can potentially contribute to the fight against bacterial infections is desperately needed. The review seeks to expose the potential contribution of aPDT and some of the non-invasive combinations.

Reviewer

This may be true, but the impact of antimicrobial PDT will likely be minimal for treatment of systemic disease.

Response

This observation is warmly acknowledged. aPDT has not proven its worth for systemic disease (anticancer PDT still has not proven its worth for metastatic disease). I share the opinion that it is unlikely to do so in the short-to-medium term.

Reviewer

The major publications are covered in this review. Whether the procedures described will have widespread utilization is unknown since many of the techniques are not generally available and in vivo efficacy is infrequently established. So while an updated review of this material can be useful, so would comments on the likelihood that techniques proposed will reach general clinical practice or whether they will be confined to a few centers.

Response

The reviewer makes a fundamental point which is accepted as a useful contribution to improvement of the review. The recommended commentary is appended after the conclusion. Although aPDT will continue to make contributions to clinical practice, some of the combinations are more likely to remain in exploration for quite some time. For example aPDT in combination with MHT will have to remain in experimental studies until biofilm eradication by magnetic hyperthermia is proven and also until the availability of MHT applicators is globalized. While the combination of aPDT with antibiotic chemotherapy is likely to continue to flourish, reaching clinical milestones with antibiotic drugs, it will still be limited to skin and shallow wound therapy. Similarly, despite the reported in-vitro synergism and efficacious in-vivo studies, the combination of PTT and aPDT is more likely to be limited to topical and shallow infections, as will CAP in combination with aPDT. However, the advancement of the SDT combination with aPDT will allow for treatment of deeper-lying and obscure disease. Nanozyme enhanced aPDT is still at its earliest phases of the proof of concept and has a long way to go and similar hurdles to overcome if it is to make a significant contribution beyond topical and shallow infections. aPDT and some of the non-invasive combinations are making contributions to various dental therapeutics and are likely to continue this trend.